# Multi-Centered Solid-Phase Quasi-Intramolecular Redox Reactions of [(Chlorido)Pentaamminecobalt(III)] Permanganate—An Easy Route to Prepare Phase Pure CoMn$_2$O$_4$ Spinel

Fernanda Paiva Franguelli [1,2], Éva Kováts [3], Zsuzsanna Czégény [1], Laura Bereczki [1], Vladimir M. Petruševski [4], Berta Barta Holló [5], Kende Attila Béres [1,6], Attila Farkas [7], Imre Miklós Szilágyi [2] and László Kótai [1,8,*]

1 Research Centre for Natural Sciences, Hungarian Academy of Sciences, Magyar Tudósok krt. 2, H-1117 Budapest, Hungary; ferpaiva@gmail.com (F.P.F.); czegeny.zsuszanna@ttk.hu (Z.C.); nagyne.bereczki.laura@ttk.hu (L.B.); beres.kende.attila@ttk.hu (K.A.B.)
2 Department of Inorganic and Analytical Chemistry, Budapest University of Technology and Economics, Műegyetem rakpart 3, H-1111 Budapest, Hungary; imre.szilagyi@mail.bme.hu
3 Wigner Research Centre for Physics (RCP), 761 Institute for Solid State Physics and Optics, Konkoly-Thege Miklós út 29-33, H-1121 Budapest, Hungary; kovats.eva@wigner.mta.hu
4 Faculty of Natural Sciences and Mathematics, Ss. Cyril and Methodius University, 1000 Skopje, North Macedonia; vladimirpetrusevski@yahoo.com
5 Department of Chemistry, Biochemistry and Environmental Protection, Faculty of Sciences, University of Novi Sad, Trg Dositeja Obradovića 3, 21000 Novi Sad, Serbia; hberta@uns.ac.rs
6 Institute of Chemistry, ELTE Eötvös Loránd University, Pázmány Péter s. 1/A, H-1117 Budapest, Hungary
7 Department of Organic Chemistry, Budapest University of Technology and Economics, Budafoki út 8, H-1111 Budapest, Hungary; farkas.attila@mail.bme.hu
8 Deuton-X Ltd., Selmeci u. 89, H-2030 Érd, Hungary
* Correspondence: kotai.laszlo@ttk.hu

**Abstract:** We synthesized and structurally characterized the previously unknown [Co(NH$_3$)$_5$Cl](MnO$_4$)$_2$ complex as the precursor of CoMn$_2$O$_4$. The complex was also deuterated, and its FT-IR, far-IR, low-temperature Raman and UV-VIS spectra were measured as well. The structure of the complex was solved by single-crystal X-ray diffraction and the 3D-hydrogen bonds were evaluated. The N-H . . . O-Mn hydrogen bonds act as redox centers to initiate a solid-phase quasi-intramolecular redox reaction even at 120 °C involving the Co(III) centers. The product is an amorphous material, which transforms into [Co(NH$_3$)$_5$Cl]Cl$_2$, NH$_4$NO$_3$, and a todorokite-like solid Co-Mn oxide on treatment with water. The insoluble residue may contain {Mn$_4^{III}$Mn$^{IV}_2$O$_{12}$}$_n^{4n-}$, {Mn$_5^{III}$Mn$^{IV}$O$_{12}$}$_n^{5n-}$ or {Mn$^{III}_6$O$_{12}$}$_n^{6n-}$ frameworks, which can embed 2 × n (Co$^{II}$ and/or Co$^{III}$) cations in their tunnels, respectively, and 4 × n ammonia ligands are coordinated to the cobalt cations. The decomposition intermediates decompose on further heating via a series of redox reactions, forming a solid Co$^{II}$M$^{III}_2$O$_4$ spinel with an average size of 16.8 nm, and gaseous N$_2$, N$_2$O and Cl$_2$. The CoMn$_2$O$_4$ prepared in this reaction has photocatalytic activity in Congo red degradation with UV light. Its activity strongly depends on the synthesis conditions, e.g., Congo red was degraded 9 and 13 times faster in the presence of CoMn$_2$O$_4$ prepared at 550 °C (in air) or 420 °C (under N$_2$), respectively.

**Keywords:** permanganate; ammine; quasi-intramolecular redox reaction; cobalt manganite catalyst; spinel; todorokite; photochemical degradation; Congo red

## 1. Introduction

Nanosized mixed transition metal-manganese oxides can be prepared by the controlled thermal decomposition of transition metal permanganate complexes that have reducing inorganic [1–6] or organic ligands [7,8]. For example, Mansouri et al. prepared a cobalt manganese oxide spinel (CoMn$_2$O$_4$) with excellent activity in the Fischer–Tropsch

fuel synthesis, by the thermal decomposition of the $[Co(NH_3)_4CO_3]MnO_4$ complex [6,9]; however, $[Co(NH_3)_4CO_3]MnO_4$ contains Co and Mn in a 1:1 atomic ratio. Thus, the decomposition product is probably a mixed $Co(Co_{0.5}Mn_{1.5})O_4$ spinel or a mixture of $CoMn_2O_4$ and $Co_3O_4$. To prepare a phase pure $CoMn_2O_4$ spinel, we attempted to prepare the $[Co(NH_3)_6]Cl(MnO_4)_2$ (**compound 1-Mn**) precursor (Co:M$n$ = 1:2), but despite the existing analogous perchlorate ($[Co(NH_3)_6]Cl(ClO_4)_2$) (**compound 1-Cl**) [10], our efforts were not successful. Alvisi [10] supposed that an unidentified chlorine-containing amminecobalt permanganate complex prepared by Klobb [11] was identical with **compound 1-Mn**, however, our repeated experiments showed that the mentioned compound was $[Co(NH_3)_6]Cl_2(MnO_4)$ (**compound 2-Mn**) (Co:M$n$ = 1:1) [11]. Moreover, we were able to produce an amminecobalt(III) permanganate complex precursor with the expected Co:M$n$ = 1:2 ratio with the isolation of the formerly unknown $[Co(NH_3)_5Cl](MnO_4)_2$ complex (**compound 3-Mn**). This precursor complex was structurally and spectroscopically characterized and heated, as a result of which it transformed into $CoMn_2O_4$ in an expected quasi-intramolecular redox reaction between the oxidizing (Co$^{III}$ and permanganate) and reducing (ammonia and chloride) components. The labels for the synthesized compounds and composites used in the paper are given in Table 1.

**Table 1.** Labels of compounds.

| Compound | Label |
|---|---|
| $[Co(NH_3)_6]Cl(MnO_4)_2$ | **1-Mn** |
| $[Co(NH_3)_6]Cl(ClO_4)_2$ | **1-Cl** |
| $[Co(NH_3)_6]Cl_2(MnO_4)$ | **2-Mn** |
| $[Co(NH_3)_5Cl](MnO_4)_2$ | **3-Mn** |
| $[Co(NH_3)_5Cl](ClO_4)_2$ | **3-Cl** |
| $[Co(NH_3)_5Cl])ReO_4)_2 \cdot nH_2O$ | **3-Re** |
| $[Co(NH_3)_5Cl]Cl_2$ | **4** |

## 2. Results and Discussion

### 2.1. Synthesis and Properties of *Compound* **3-Mn**

[Chlorido(pentaammine)cobalt(III)] permanganate (**compound 3-Mn**) has not been prepared previously, although Krestov and Yatsimirksii [12] calculated its thermodynamical properties. The analogous perchlorate compound ($Co(NH_3)_5Cl)(ClO_4)_2$ (**compound 3-Cl**) is known [13] and can easily be prepared by heating the [(aqua)(pentaammine)cobalt(III)] triperchlorate solution in 1 M HCl. Unfortunately, this reaction route cannot be used to prepare **compound 3-Mn** due to the spontaneous reaction between the permanganate ions and the HCl solution (chlorine evolution and Mn$^{II}$ formation). Therefore, a chloride ion has to be introduced into the inner coordination sphere of the cobalt in the absence of a permanganate ion—for example, with the preparation of $[Co(NH_3)_5Cl]Cl_2$ (**compound 4**) [14]—and the outer sphere chloride ions have to be exchanged with permanganate ions, such as occurred with the perrhenate ions in the preparation of **compound 3-Re** [15,16]. A possible reaction route is the reaction of $[Co(NH_3)_5Cl]SO_4$ [14] with barium permanganate [17–20]; however, this method was not used due to the relatively low solubility of **compound 3-Mn** in water that causes problems in the separation of **compound 3-M** from the insoluble barium sulfate by-product. The reaction of **compound 4** and an excess of $NaMnO_4$ in a 40% aq. solution, with subsequent cooling to +1 °C resulted in the formation of the **compound 3-Mn** as a reddish-purple precipitate with a 57% yield. Its powder X-ray diffractogram is given in Figure S1. Its solubility in water was 0.0089 M (3.71 g/L) and 0.0225 M (9.39 g/L) at 0 °C and 25 °C, respectively. It is insoluble in aliphatic hydrocarbons and such polar organic solvents like chloroform, dichloromethane, or benzene, but easily soluble in DMF, and decomposes in DMSO.

### 2.2. Structure of **Compound 3-Mn**

Dark violet single crystals of **compound 3-Mn** were selected from the precipitate formed during the synthesis. The powder X-ray diffractogram calculated from the SXRD data agreed very well with the experimentally determined powder X-ray diffractogram (Figure S1). Crystallographic data, details of the structure and the refinement results are listed in Table 2, together with the data obtained for the analog perrhenate complex (**compound 3-Re**) [15]. The structural features of **compound 3-Mn** are shown in Figures 1–3. The detailed structural parameters, bond lengths and angles can be found in Tables S1–S3.

**Table 2.** Crystal data of **compound 3-M**.

| Empirical Formula | $[Co(NH_3)_5Cl](MnO_4)_2$ | $[Co(NH_3)_5Cl](ReO_4)_2 \cdot nH_2O$ [15] |
|---|---|---|
| Formula Weight | 417.4101 g·mol$^{-1}$ | 679.95 g·mol$^{-1}$ |
| Crystal System | Orthorhombic | Orthorhombic |
| Space Group | $Cmc2_1$ | $Cmc2_1$ |
| Unit cell dimensions, Å | a = 14.2753 (7) | a = 14.9446 (3) |
| | b = 14.2816 (6) | b = 14.6562 (4) |
| | c = 12.2342 (5); | c = 12.2434 (4) |
| Z | 8 | 8 |
| Density (calcd.) (g·cm$^{-3}$) | 2.216 | 3.368 |
| Temperature (K) | 298 | 293 |
| Volume (Å$^3$) | 2494.24 (19) | 2681.68 (13) |
| R factor (%) | 3.67 | 2.35 |

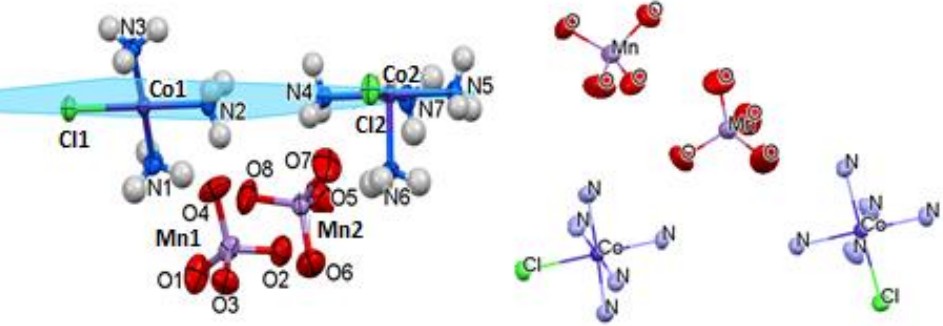

**Figure 1.** Content of the asymmetric unit of $[Co(NH_3)_5Cl](MnO_4)_2$ containing two half complex cations and two anions (the light blue plane is a mirror plane cutting the complex cations into two halves; thermal ellipsoids are drawn at a 50% probability level).

**Compound 3-M** is orthorhombic. Its space group is $Cmc2_1$ (Nr. 36). The coordination around the Co$^{3+}$ cation is octahedral. The asymmetric unit contains two half complex $[Co(NH_3)_5Cl]^{2+}$ cations and two permanganate anions (Figure 1). The central cobalt ions of the complex cations, the coordinated chloride anions, and some of the ammonia ligands sit on a mirror plane and therefore have half-site occupancy, so that the hydrogens of these ammonia molecules are disordered over two mirrored positions. The structural features of **compound 3-Mn** are shown in Figures 1–3.

The axial Co-N bond distance (Co-N1 = 1.955 (6) Å) was shorter in cation A than the equatorial bond distances (1.964(4)–1.965(4) Å), whereas in cation B, the axial Co-N distance (Co-N7 = 1.964 (6) Å) was longer than the equatorial ones (1.950(4)–1.960 (6) Å). The Co-Cl bonds were almost equal (2.258 (2) and 2.255 (2) Å in cation A and B, respectively). The Co-Cl distances in the analogous perrhenate complex (**Compound 3-Re**) were found to be 2.270 (2) and 2.260 (2) Å, whereas the Co-N distances varied between 1.960 (5)–1.972 (6) and 1.948 (7)–1.988 (4) Å, respectively. Layers of the complex cations in the bc plane were embedded within permanganate layers (Figure 2).

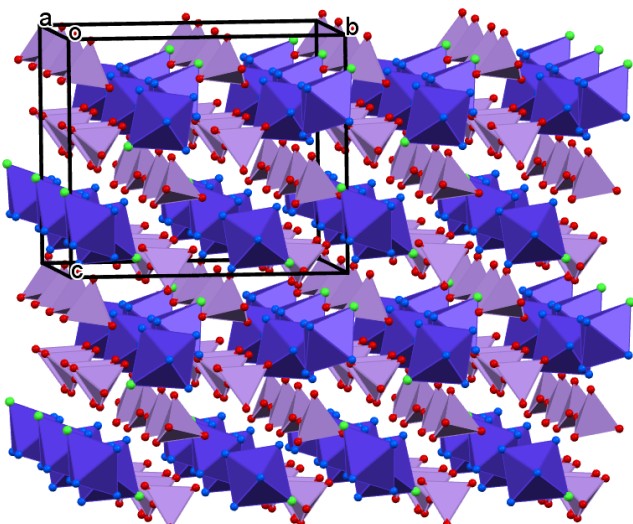

**Figure 2.** Parallel $[Co(NH_3)_5Cl]^{2+}$ (dark violet octahedra) and $MnO_4^-$ (light violet tetrahedra) layers in the bc plane (view in the direction of the c axis).

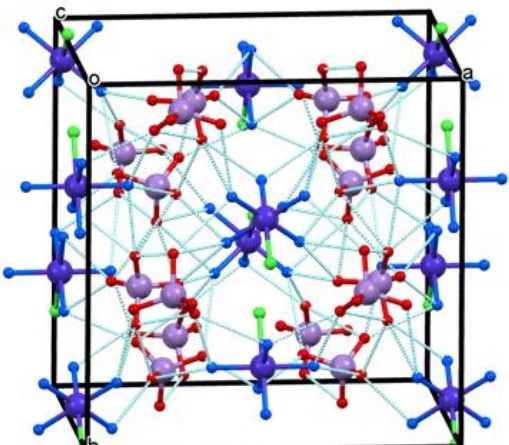

**Figure 3.** Packing arrangement of $[Co(NH_3)_5Cl](MnO_4)_2$ and hydrogen bonds in light blue (hanging contacts are omitted for clarity) between the ions.

The $[Co(NH_3)_5Cl]^{2+}$ cation has five hydrogen donor groups and one hydrogen acceptor group. The arrangement of the cations in the crystal lattice is such that all ammonia hydrogens can find at least one acceptor (chloride or permanganate oxygen) (Figure 3, Table S3).

The strengths of N-H . . . O and N-H . . . Cl hydrogen bonds in **compound 3-Mn** were comparable with the hydrogen bond strengths of the analogous perrhenate complex (**compound 3-Re**) (N . . . O distances were between 2.86–3.23 Å and 2.91–3.29 Å, whereas N . . . Cl distances were between 2.98–3.60 Å and 3.33–3.53 Å for **compound 3-Mn** and **compound 3-Re**, respectively [15,16]).

### 2.3. Spectroscopic Properties of *Compound* 3-*Mn*

We analyzed the vibrational spectroscopic modes of **compound 3-Mn** by factor group analysis, considering the structure of $[Co(NH_3)_5Cl](MnO_4)_2$ to be composed of a Co–Cl group, five $NH_3$ molecules coordinated to Co (four and three crystallographic types at the special and trivial positions $C_s$ and $C_1$, respectively), and two types of isolated $MnO_4^-$ anions. The **compound 3-Mn** is orthorhombic ($Cmc2_1$, Z = 8), consisting of two types of positions only: general (with trivial symmetry, 1) and special (with symmetry *m*, *yz* plane). The unit cell is base-centered. Thus, the primitive cell contains only four motifs, meaning

$Z' = 4$ for the "spectroscopic cell". This means that there are two quartets of permanganate anions, each at the position of trivial symmetry. The internal of the $NH_3$ molecules at the special and trivial symmetry positions ($C_s$ and $C_1$) can be seen in Figure 4. The internal and external vibrations of the permanganate ions and Co-Cl groups (molecular symmetry $C_{\infty v}$), or external vibrations for the $NH_3$ ligands in **compound 3-Mn**, are presented by Figures S2–S4.

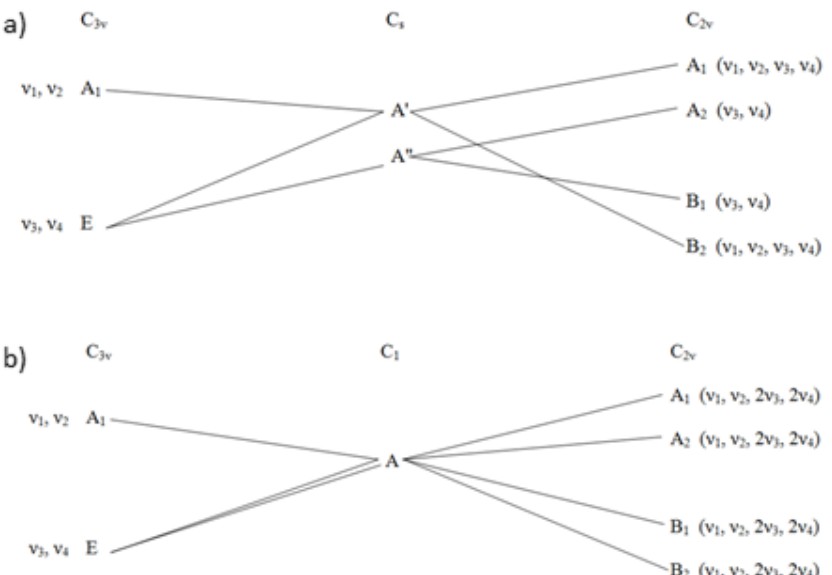

**Figure 4. (a)** Internal vibrations (four crystallographic types) of $NH_3$ molecules at special symmetry positions $C_s$ in **compound 3-Mn**; **(b)** Internal vibrations (three different crystallographic types) of $NH_3$ molecules at positions of trivial symmetry, $C_1$, in **compound 3-Mn**. $\nu_1$—symmetric stretch.; $\nu_2$—symmetric bend.; $\nu_3$—antisymmetric stretch; $\nu_4$—antisymmetric bend.

### 2.3.1. Vibrational Modes of the Permanganate Anion

The vibrational modes of the permanganate ion were expected to appear in both the IR and Raman spectra as two series of singlet symmetric ($\nu_1$), triplet antisymmetric ($\nu_3$), doublet symmetric ($\nu_2$) and triplet antisymmetric ($\nu_4$) ones. The IR-forbidden $\nu_1$ and $\nu_2$ under $T_d$ became IR active due to symmetry lowering ($C_{2v}$) (Figure 5).

The IR, Raman and far-IR spectra of **compound 3-Mn** are given in Figures 4, 5 and S5. The low-temperature Raman spectral range contains the permanganate and $[CoN_5Cl]^{2+}$ skeleton vibrations of **compound 3-Mn,** which can be seen in Figure 6. The IR and Raman data assignments can be seen in Tables 3–5.

The low-temperature Raman spectra for **compound 3-Mn** were recorded in the range of 1000–200 cm$^{-1}$ (Figure 6).

Despite the forbidden nature of $\nu_s$(Mn-O) and $\delta_s$(Mn-O) modes in $T_d$, 2 and 2 × 2, weak singlet IR bands, respectively, were expected to appear due to the distortion of the $MnO_4^-$ symmetry ($C_{2v}$) and the presence of two sets of permanganate ions in the structure; however, the IR spectrum contained one singlet band at 841 and one at 352 cm$^{-1}$, assigned as $\nu_s$(Mn-O) and $\delta_s$(MnO) modes. The band at 841 cm$^{-1}$ had a shoulder (Figure 5). The $\nu_s$(Mn-O) mode was expected and found to be an intensive band in the Raman spectrum without splitting at room temperature and as two singlets at 123 K, whereas the $\nu_s$(Mn-O) appeared as a singlet at room temperature and as a singlet with two shoulders in the low-temperature Raman spectra. The relatively high IR intensity of the $\nu_s$(Mn-O) band can be attributed to the occurrence of a vibrational mode of the cation core $[Co(NH_3)_5Cl]^{2+}$ located around this wavenumber and coinciding with the $\nu_s$(Mn-O) band. Abbas [21] and Najar [22] assigned a band around ~844 cm$^{-1}$ to $\nu$(Co-Cl), whereas Sacconi [23] assigned a band around 849 to $\rho(NH_3)$ in the IR spectrum of $[Co(NH_3)_5Cl]Cl_2$ (**compound 4**). Thus,

one of these modes might coincide with the band of $\nu_s$(Mn-O) and play a role in the appearance of an intensive mixed IR band (a consequence of vibrational resonance). A detailed study of the skeletal vibrational modes [24–26] (see below), including the $\nu$(Co-Cl) mode, showed that it was located in the far-IR range and not around 844 cm$^{-1}$. Thus, the coinciding component may only be the $\rho$(NH$_3$) mode. The low-temperature (123 K) Raman spectrum of **compound 3-Mn** showed two bands at 849 and 839 cm$^{-1}$ (Figure 6), with similar intensities. The $\rho$(NH$_3$) mode could not be seen in the Raman spectrum [25] but appeared in the IR spectrum of **compound 4** [21,25]. Thus, the splitting of the singlet Raman band into two components can only be attributed to the appearance of the singlets of two kinds of permanganate ions located in the structure of **compound 3-Mn**. The lack of $\rho$(NH$_3$) band in the Raman spectrum and its role in the increase in the intensity of the mixed $\nu_s$(Mn-O) + $\rho$(NH$_3$) IR band were confirmed by the deuteration of **compound 3-Mn** as well. The IR and Raman spectra of the deuterated **compound 3-Mn** (Figures S6–S8) showed that the $\rho$(NH$_3$) component of the mixed IR band was decomposed (the shape of the band at 841 cm$^{-1}$ was changed and a new band appeared at 646 cm$^{-1}$ ($\rho$(ND$_3$)), whereas the $\nu_s$(Mn-O) band structure in the LT-Raman spectrum was not changed at all by deuteration. The antisymmetric stretching mode ($\nu_{as}$(Mn-O) was a strong band in the IR and a weak one in the Raman spectra at room temperature (Table 3). The low-temperature Raman study (123 K, Figure 6) showed splitting of the band located around ~900 cm$^{-1}$ into six components, which corresponded to the two sets of triplet $\nu_{as}$(Mn-O)(F$_2$) modes. There were no coinciding cationic modes in this spectral range.

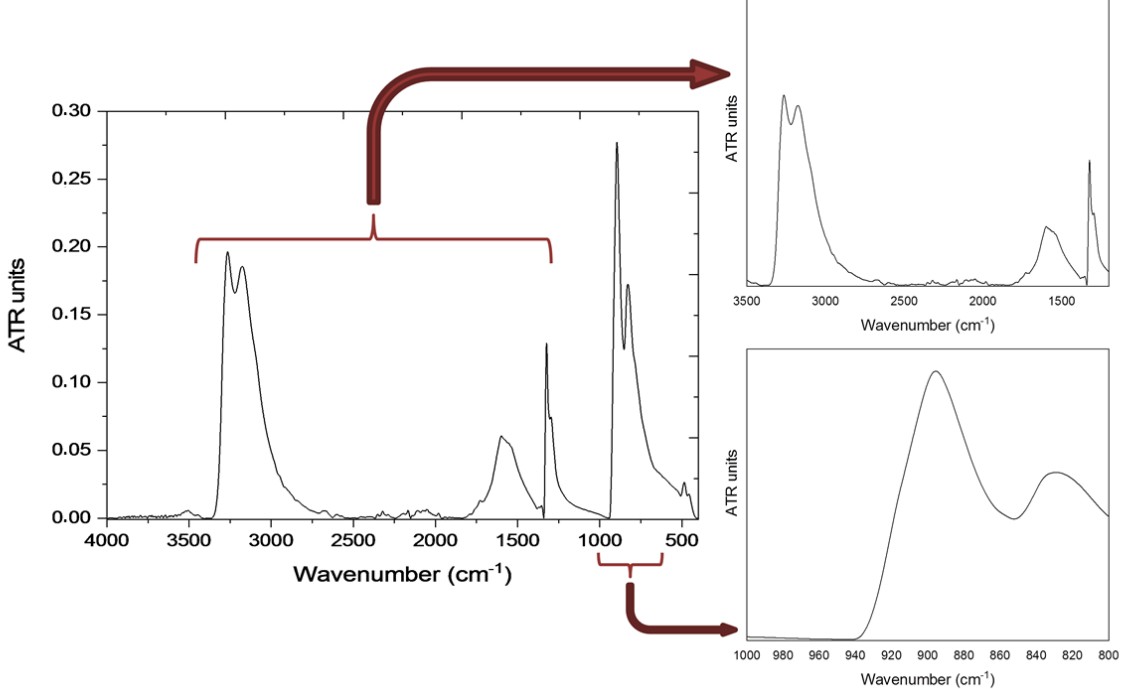

**Figure 5.** IR spectrum of **compound 3-Mn** in the range of 4000–400 cm$^{-1}$ at room temperature.

The symmetric deformation mode ($\delta_s$(Mn-O) was double degenerated, and according to this, 2 × 2 = 4 bands were expected both in the far-IR and Raman spectra. The far-IR and the LT-Raman spectra contained one and two bands at 381 and 386/396 cm$^{-1}$, respectively. The intensities of the two bands in the LT-Raman spectrum were almost equal, and the asymmetric shape of the bands showed that these consisted of more than one component. Since $\delta_{s\,s}$(Mn-O) is a double degenerated mode, and there are two sets of permanganate ions in the structure, the two bands probably contain 2 components each. The wide band found at room temperature Raman spectra (Figure 6) split at 123 K. This may be attributed to the appearance of the two sets of permanganate ion modes and due to the splitting of the

E symmetry mode. The antisymmetric $\nu_{as}$(Mn-O) mode appeared in both the far-IR and the LT-Raman spectra around 349 (sh) and 346 cm$^{-1}$ (Figure 6 and Figure S5), respectively.

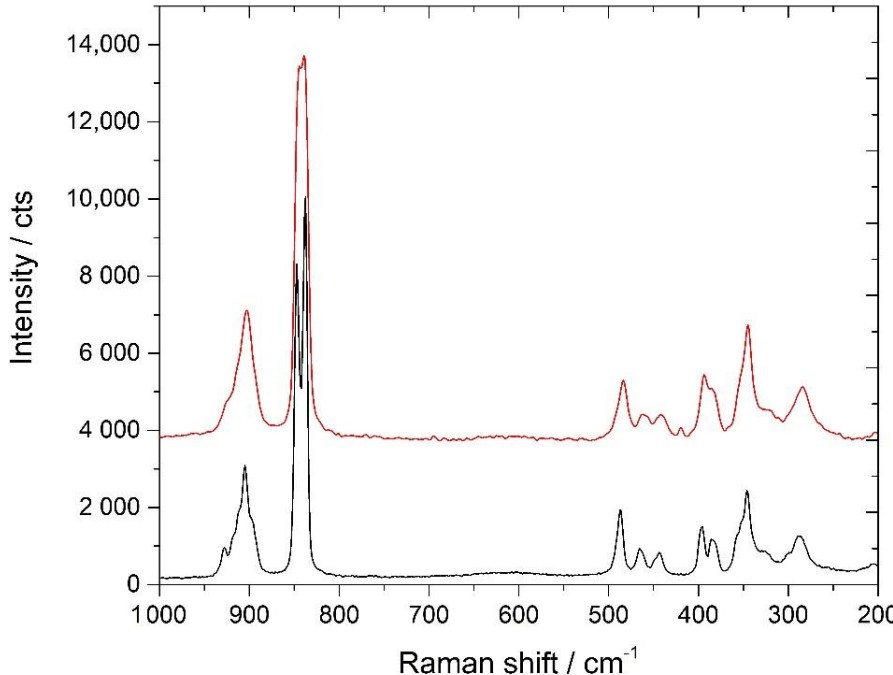

**Figure 6.** Raman spectra of **compound 3-Mn** at 298 (red) and 123 K (black) in the range of 1000–200 cm$^{-1}$.

**Table 3.** Assignments of permanganate ion vibrational modes in the IR and Raman spectra of **compound 3-Mn**.

| Band/Assignation | Wavenumber/Raman Shift (cm$^{-1}$) | | |
|---|---|---|---|
| | IR | Raman (785 nm) | |
| | | 298 K | 123 K |
| $\nu_1$ (MnO), $\nu_{as}$ (A) * | 829 | 841 | 848, 838 |
| $\nu_3$ (MnO), $\nu_{as}$ (F$_2$) | 895 | 903 | 928, 918sh, 913sh, 905sh, 899sh |
| $\nu_4$ (MnO), $\delta_{as}$ (F$_2$) | 379 | 389 | 396, 386, 383 |
| $\nu_2$ (MnO), $\delta_s$ (E) | 352 | 346 | 352sh, 358sh, 346 |

* Mixed band of $\nu_1$ (Mn-O) and $\rho$(NH$_3$).

### 2.3.2. Ligand Vibrations

The correlation analysis of the cationic part of **compound 3-Mn** showed seven (4 in C$_s$ and 3 in C$_1$) and two sets of vibrational modes belonging to ammonia molecules (Figure 4) and Co-Cl groups (Figure S3). The Co-N and Co-Cl modes were evaluated as skeletal modes, and the assignation of the vibrational modes of the ammonia ligands in **compound 3-Mn** and its deuterated form (Figures S6–S8) are given in Table 4.

The differences in the geometrical (axial and equatorial) and the seven crystallographic positions of ammonia molecules resulted in poorly resolved complex band systems for all kinds of N-H modes. The symmetric deformation mode $\nu_s$ (N-H) of **compound 3-Mn** showed two main types of ammonia ligated with different strengths to the central Co$^{III}$-ion. A relative bond strength parameter ($\varepsilon$) for the ammonia molecules in ammine complexes was defined by Grinberg [27,28]. The parameter ($\varepsilon$) was found to be 0.88 and 0.80 as the two types of coordinated ammonia molecules gave two different $\delta_s$ (N-H) bands in the IR spectrum of **compound 3-Mn**. Thus, the bond strength difference between the two main types of coordinated ammonia molecules in **compound 3-Mn** is ~10%. This difference may

be attributed to the change in the strength of the Co-N bond (apical or equatorial position); however, the electron density change caused by this difference might be attributed to the difference between the hydrogen bond strength of each ammonia molecule, which can play a crucial role in the value of the Grinberg parameter [2,28].

**Table 4.** The IR and Raman spectra of the ammonia ligand in $[Co(NH_3)_5Cl](MnO_4)_2$ (**compound 3-Mn**) and its deuterated form.

| Band/Assignation | Wavenumber (cm$^{-1}$) | | | |
| --- | --- | --- | --- | --- |
| | Compound 3-Mn | | Deuterated Compound 3-Mn | |
| | IR, 298 K | Raman (785 nm), 123 K | IR, 298 K | Raman (785 nm), 123 K |
| $\nu_{as}$ (HNH)(DND) | 3293 | - | 2444 | - |
| $\nu_s$ (HNH)(DND) | 3270 | - | 2319 | - |
| $\delta_{as}$(HNH)(DND) | 1607 | - | 1153 | - |
| $\delta_s$(HNH)(DND) | 1324, 1294 (m) | 1352, 1311 (vw) | 1015 | - |
| $\rho$ (NH$_3$)(ND$_3$) | 829 (s) | - | 646 | - |

The positions of N-H stretching and deformation N-H modes in the IR spectrum of **compound 3-Mn** did not give unambiguous information about the presence and strength of hydrogen bonds in the **compound 3-Mn**. The $\rho$ (NH$_3$) rocking mode is the most sensitive vibrational mode to detect the presence of hydrogen bonds in ammonia complexes [23], but the $\rho$(NH$_3$) in the IR spectrum of the **compound 3-Mn** was a mixed band with $\nu_s$ (Mn-O). Therefore, we prepared the deuterated **compound 3-Mn** and determined the $\rho$ (ND$_3$) band position (646 cm$^{-1}$). Using the $\rho$(NH$_3$)/$\rho$(ND$_3$) wavenumber ratio found for $[Co(NH_3)_5X]X_2$ (1.267) compounds [23], we calculated the position of $\rho$(NH$_3$) for **compound 3-Mn** and found it to be 818 cm$^{-1}$. This showed that the strength of the hydrogen bond in **compound 3-Mn** is weaker than in $[Co(NH_3)_5Cl]Cl_2$ ($\rho$(NH$_3$) = 849 cm$^{-1}$) and is between those found in $[Co(NH_3)_5Br]Br_2$ ($\rho$(NH$_3$) = 830 cm$^{-1}$) and $[Co(NH_3)_5I]I_2$ ($\rho$(NH$_3$) = 810 cm$^{-1}$) [23].

### 2.3.3. Skeletal Vibrations

The skeletal vibrations were evaluated by the normal coordinate analysis of the $[Co(NH_3)_5Cl]^{2+}$ ion [24–26]. For the skeleton $[MN_5X]^{2+}$, assuming $C_{4v}$ symmetry, 11 normal modes were expected. The $A_1$ and E species were IR and Raman active, whereas the $B_1$ and $B_2$ species were only Raman active. The normal modes and band assignments for **compound 3-M** and its deuterated form are given in Table 5.

Based on the correlation analysis results (Figure S3), two sets of $\nu$(Co-Cl) modes were expected and found at 278/272sh and 288/285 cm$^{-1}$ in the far-IR (Figure S5) and LT-Raman spectra (Figure 6), respectively. The slight shifts of these bands on deuteration (**compound 3-Mn** and its deuterated form) (average $\nu_H/\nu_D$ = ~1.04) can be attributed to the coupling of the $\nu$(Co-Cl) mode with other skeletal vibrational modes. The intensity of some bands was very weak but based on the data of normal coordinate analysis of the $[Co(NH_3)_5Cl]^{2+}$ and $[Co(ND_3)_5Cl]^{2+}$ skeletons, all bands were successfully assigned (Table 5).

### 2.4. UV-VIS Spectroscopy

The solid-phase UV-VIS spectrum of **compound 3-Mn** was recorded at room temperature, but the spectrum contained strongly overlapping bands both in the UV and visible region (Figure S9). The band at 217 nm may have belonged to both the CT band of the complex cation (electron transfer is from chlorine to the $d_{z2}$ orbital of cobalt (LMCT)) and the $^1A_1$-$^1T_2$ ($t_1$-$4t_2$) transition of a permanganate ion as well. The Cl to Co band position for **compound 4** and the $^1A_1$-$^1T_2$ ($t_1$-$4t_2$) transition for KMnO$_4$ were found at 227 nm [8,29,30]. A wide band was observed with maximums at 249 and 266, which may have belonged to

the $^1A_1$-$^1T_2$ (3t$_2$-2e) and $^1T_{2g}$-$^1A_{1g}$ transition of the permanganate ion and the octahedral $[Co(NH_3)_5Cl]^{2+}$ cation [29,30], respectively. The analogous bands were found at 259 and 274 nm for the KMnO$_4$ and **compound 4**, respectively [8,29,30]. Due to the purple color of the complex cation and the permanganate anion in **compound 3-Mn**, the visible region contained a very intensive band system between 500 and 600 nm. The permanganate $^1A_1$-$^1T_2$ (t$_1$-2e) band for the KMnO$_4$ was found between 500 and 562 nm [8], and the doublet of the $^1T_{1g}$-$^1A_{1g}$ transition of the $[Co(NH_3)_5Cl]_2SO_4$ was found at 470 and 547 nm [29]. The maximums at 570 (permanganate) and 546 nm (complex cation) for **compound 3-Mn** probably belonged to these transitions. No split of the $^1T_{1g}$-$^1A_{1g}$ transition of the complex cation due to the difference between the contribution of the chloride ion (1Cl + 3NH$_3$) and ammonia (4NH$_3$) coordination was observed. The band at 685 nm probably belonged to the $^1A_1$-$^1T_1$(t$_1$-2e) transition of permanganate ion. This transition was found at 720 nm for the KMnO$_4$ and 710 nm for the $[Agpy_2]MnO_4$ [8].

**Table 5.** Skeletal vibrational mode assignments in the IR and Raman spectra of **compound 3-Mn** and its deuterated form.

| Species | Mode | Assignation | Compound 3-Mn | | Deuterated Compound 3-Mn | |
|---|---|---|---|---|---|---|
| | | | IR (298 K) | Raman (123 K) | IR (298 K) | Raman (123 K) |
| A$_1$ (R IR) | $\nu_1$ | $\nu$ (Co-Cl) | 278, 272sh | 288, 285 | 264 | 265 |
| | $\nu_2$ | $\nu_s$(CoN$_4$) | 458 | 465 | 451 | 456 |
| | $\nu_3$ | $\nu$ (Co-N)axial | 488 | 487 | 483 | 489 |
| | $\nu_4$ | $\pi$ (Co-N$_4$) | 199 | 189sh | 185 | 184 |
| B$_1$ (R) | $\nu_5$ | $\nu_{as}$ (CoN$_4$) | - | 443 | - | 448 |
| | $\nu_6$ | $\pi$ (Co-N$_4$) | - | 193sh | - | 193sh |
| B$_2$ (R) | $\nu_7$ | $\delta$ (NCoN) | - | 327 | - | 327 |
| E (R, IR) | $\nu_8$ | $\nu$ (Co-N)axial | 505, 493 sh | 496 sh | 483 | 505 |
| | $\nu_9$ | $\delta$ (NCoN or CoN$_4$ wag) | 320 | 329 | 344 sh | 347 |
| | $\nu_{10}$ | $\delta$ (NCoCl or CoN$_4$ rock) | 200 | 205 | 199 sh | 203 |
| | $\nu_{11}$ | $\delta$ (NCoN or CoN$_4$ in-plane blend) | 259sh | 265 | 264 | 265 |

### 2.5. Thermal Decomposition of *Compound 3-Mn*

The thermal decomposition of [(chlorido)pentaamminecobalt(III)] permanganate (**compound 3-Mn**) was studied in an oxidative and inert atmosphere. There were no significant differences between the decomposition mechanisms in argon and air. The decomposition processes were somewhat more intensive in air than in argon, but the characteristic temperatures were almost the same. Based on the mass losses, powder XRD and the Co:M$n$ = 1:2 stoichiometry in **compound 3-M**, the final decomposition product above 400 °C was an oxide phase with the overall formula of CoMn$_2$O$_4$, with an average size of 16.8 nm (identified by powder XRD, and size determined by the Scherrer method (Figure 7)).

The thermal decomposition of **compound 3-Mn** started at 121 °C, onset (Figures S11 and S12). The mass loss in the first decomposition step was ~4.9%, which was somewhat more than the mass percent of one molecule of NH$_3$ (4.08%).

The IR study supported the notion that the ammonia molecules were bonded with various strengths ($\varepsilon$ = 0.80 and 0.88, see above), which predicted a possible stepwise ammonia loss. The TG-MS study, however, unambiguously showed that in the first decomposition step, the leaving product was water ($m/z$ = 18, H$_2$O$^+$) and not ammonia ($m/z$ = 17, NH$_3^+$) (Figure 8 and Figure S11). The shoulder at around 125 °C on the ion curve $m/z$ = 17 corresponded to the HO$^+$ fragment ion of water. Moreover, the ammonia had a significant $m/z$ = 16 peak but the ion intensity curve of $m/z$ = 16 did not have any peak or shoulder at around 125 °C. Neither N$_2$ nor NO$_x$ compounds were formed in this step. **Compound 3-Mn** is anhydrous. Thus, water may only be the reaction product of a solid-phase quasi-intramolecular redox reaction between ammonia as the hydrogen source

and a species containing oxygen. Due to the inert atmosphere, the oxygen source may only be the permanganate ion. Thus, this redox reaction was centered on ammonia ligands and permanganate ions.

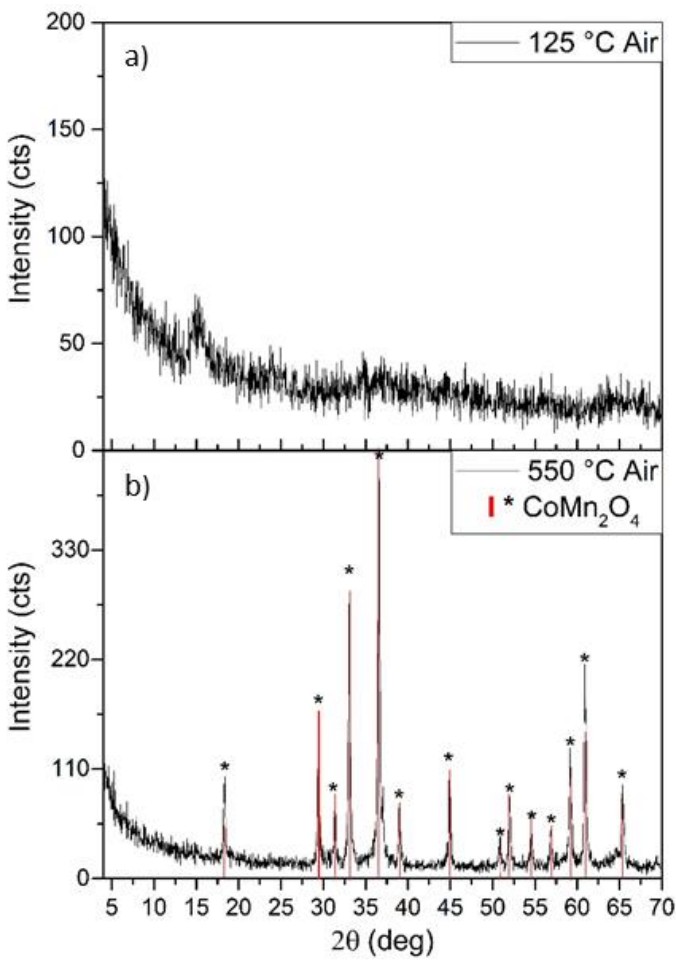

**Figure 7.** Powder X-ray diffractogram of the decomposition products of **compound 3 Mn** at 125 (**a**) and 550 °C (**b**) in 1 h in the air atmosphere.

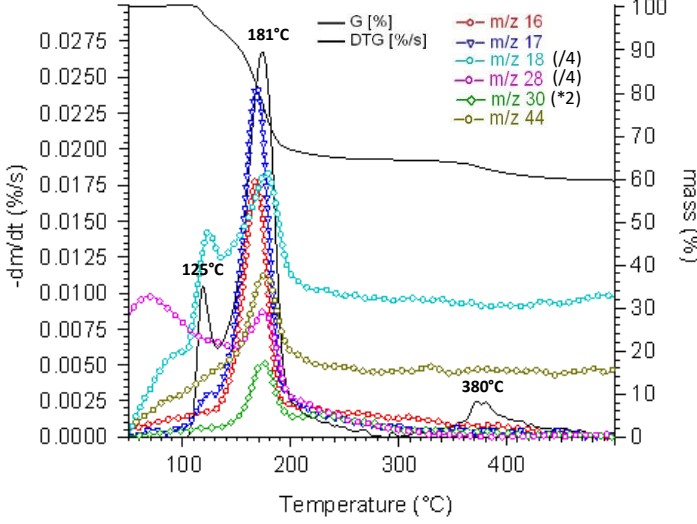

**Figure 8.** Thermal decomposition curves of **compound 3-Mn** in an argon atmosphere.

The IR spectra of the decomposition product formed from **compound 3-Mn** at 125 °C (Figure 9) showed that the permanganate ion IR bands almost completely disappeared. There were no signs of manganate (VI) or manganate (V) ions [17,19]. Thus, manganese valence in the decomposition product may only be +2, +3, or +4. The release of one water molecule left seven oxygen atoms remaining from the eight oxygen atoms of the two permanganate ions in **compound 3-M**, distributed among the possible redox products. The primary decomposition product was X-ray amorphous, but its IR (Figure 9) and far-IR spectra (Figure S12) showed the presence of an ammonium ion (shoulder at ~1680 cm$^{-1}$ ($\delta_{as}$(N-H)), $\nu_2 + \nu_4$ combination band at ~2840 cm$^{-1}$) and nitrate ions ($\nu_{as}$(N-O) ~1370 cm$^{-1}$). The N-H bands belonging to $[Co^{II}(NH_3)_4]^{2+}$ ions were expected to appear around the same wavenumbers as the $NH_3$ bands of **compound 4** [31]. The decomposition intermediate showed the band characteristics for todorokite and related mixed manganese oxide family members as containing $Mn^{III}/Mn^{IV}$ species [32].

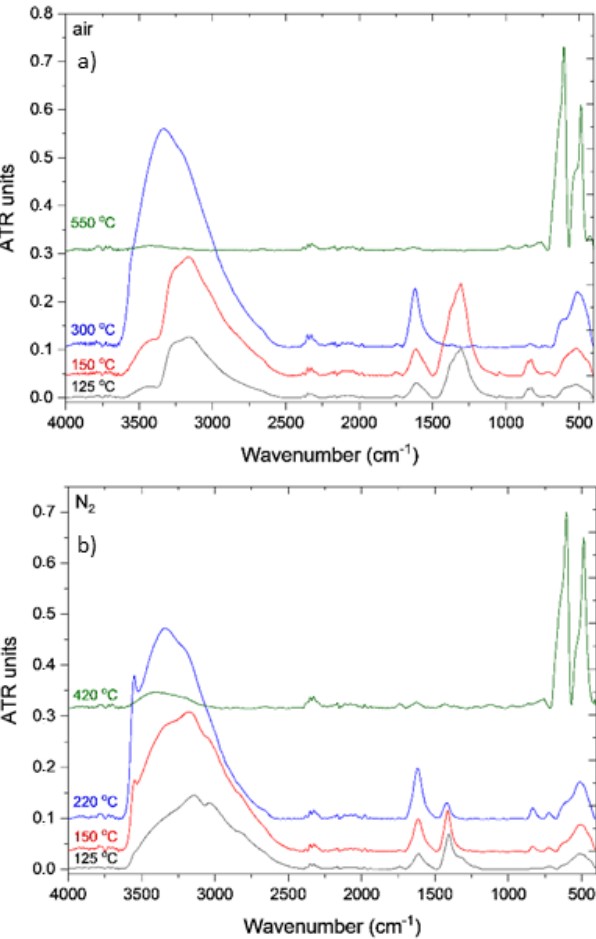

**Figure 9.** IR spectra of thermal decomposition intermediates of **compound 3-Mn** formed under air (**a**) and nitrogen (**b**) atmospheres.

The first thermal decomposition step of the **compound 3-Mn** was also studied in refluxing toluene. The boiling temperature of toluene (110 °C) limits the maximal decomposition reaction temperature by absorbing the evolved reaction heat via the evaporation of toluene, which prevents overheating by the heat of the reaction. The solid phase formed in the thermal decomposition of **compound 3-Mn** under refluxing toluene in 1 h consisted of amorphous phases (Figure 10), which could be crystallized by heating above 400 °C with the formation of $CoMn_2O_4$ and gaseous ($N_2$, $H_2O$, $NH_3$, $NH_4Cl$) products. The crystallite size of the $CoMn_2O_4$ depended on the heating time and temperature, similarly to other spinels prepared from ammine complexes of transition metal permanganates [1,3–5].

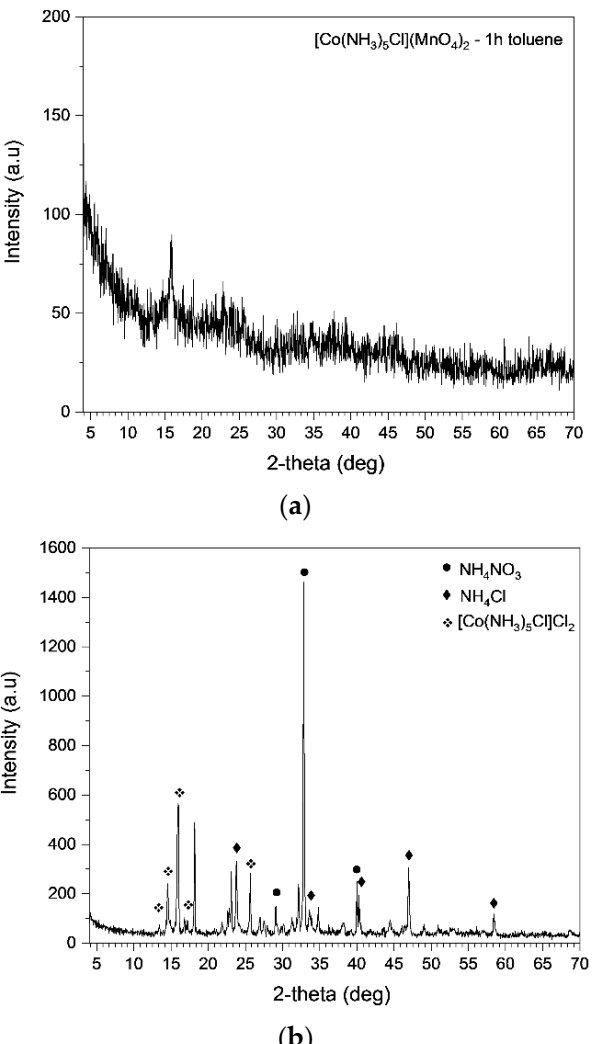

**Figure 10.** (**a**) XRD of the decomposition product of **compound 3-Mn** after 1 h reflux in toluene and (**b**) XRD of the dried aqueous extract of the thermal decomposition product formed under toluene.

Separating the amorphous reaction product into water-soluble and insoluble (oxide) parts and evaporating the aqueous leachate, we assigned the water-soluble components by powder XRD and IR as NH4NO3, [Co(NH3)5Cl]Cl2 (compound 4) and a small amount of NH4Cl (Figure 10). The NH4Cl was probably formed as a hydrolysis product of compound 4.

It is well known that ammonium nitrate forms in the solid-phase quasi-intramolecular redox reaction of the ammonia complexes of divalent metal permanganates with the formation of solid Mn (III) type spinel compounds [1–5]:

$$2NH_3 + 2MnO_4{}^- = NH_4NO_3 + H_2O + \{Mn_2O_4\}^{2-}$$

Neutralizing one charge of the trivalent cobalt in **compound 3-Mn** with a chloride ion results in a formal "divalent complex cation", and an analog reaction of this "divalent cation" can explain the formation of one mol of water and $NH_4NO_3$. The residual solid phase can be characterized with a formal $\{CoClMn_2O_4 + 3NH_3\}$ composition. The material, valence, and charge balances for chlorine, nitrogen, hydrogen, and oxygen, together with the amount of the evolved water, and the lack of ammonia and oxygen evolution, resulted in the following reaction in the first decomposition step:

$$3[Co(NH_3)_5Cl](MnO_4)_2 = [Co(NH_3)_5Cl]Cl_2 + 3H_2O + 3NH_4NO_3 + \{Co_2(NH_3)_4Mn_6O_{12}\}$$

Since no gaseous ammonia was evolved, the ammonia molecules which were not oxidized to nitrate or protonated to ammonium ions, or were not coordinated in **compound 4**, could be fixed in a coordinated form in the cobalt and manganese-containing residual oxide phase. Depending on the valences of the Co ions ($2 \times III$, $2 \times II$ or $II + III$), the charge of Co cations can neutralize $\{Mn_6O_{12}\}^{n-}$ units with six, four, or five negative charges, thus, according to this, the average valence of manganese was 18/6 ($6Mn^{III}$), 20/6 ($4Mn^{III}$ and $2Mn^{IV}$), or 21/6 ($3Mn^{III}$ and $3Mn^{IV}$), respectively. Although there was no sign of ammonia oxidation with $Co^{III}$ in this decomposition step (the lack of $N_2$ formation confirmed it [33]), the $Co^{II}$ and $Mn^{IV}$ centers in the oxide phase might have been formed due to the valence distribution between the $Co^{III}$ and $Mn^{III}$. This is a hidden solid-phase quasi-intramolecular redox reaction without mass loss and the formation of other assignable redox products.

The typical coordination number of $Co^{III}$ in its ammonia complexes is 6, and Co(II) forms stable tetraammine complexes. Thus, the stoichiometry of $Co:NH_3$ suggested that one cobalt might be expected to be divalent and complexed with four ammonia molecules. Furthermore, the composition and IR spectrum of the decomposition product (Figures 8 and S11) and the possible charges of the $\{Mn_6O_{12}\}^{n-}$ ($n$ may be 4, 5 or 6) unit strongly suggest that $[Co_2(NH_3)_4Mn_6O_{12}]$ belongs to todorokite or a similar type of manganese oxide family [34]. Todorokite is constructed of triple chains of $MnO_6$ octahedra forming large tunnels with square cross-sections [34], which can incorporate as large cations as $[Co(NH_3)_n]^{2+/3+}$ ($n$ = 2, 4 or 6) and free $Co^{II}/^{III}$ cations as well.

XPS could not directly determine the $Co^{II}/^{III}$ and $Mn^{II}/^{III}$ ratio due to the ammonia loss of $[Co_2(NH_3)_4Mn_6O_{12}]$ in a high vacuum, which led to chemical changes in the studied sample. Therefore, future $^{57}Co$ Mössbauer studies are planned to investigate the $[Co_2(NH_3)_4Mn_6O_{12}]$ phase in more detail.

The first thermal decomposition step of **compound 3-Mn** was exothermic in both an oxidative and an inert atmosphere. The reaction heats were found to be 475.85 and 479.95 kJ/mol by DSC (Figure S14a,b) in pure $O_2$ and $N_2$, respectively. Thus, the outer oxygen source did not play any role in the first step of the decomposition process.

In the second decomposition step, the mass loss was ~28.6%, and above 300 °C, a slight mass loss of 6.5% was observed. The residual mass at 500 °C was 57.7%, which was somewhat more than the mass percent of the $CoMn_2O_4$ spinel (55.77%) (Figure S10).

The TG-MS curves showed simultaneous $N_2O$ and $H_2O$ formation ($m/z$ = 44 and 18, respectively), which is typical in the thermal decomposition of ammonium nitrate:

$$NH_4NO_3 = N_2O + 2H_2O$$

Based on the $N_2O/NO$ ($m/z$ = 44 and 30, respectively) peak intensity ratios, the formation of NO may be attributed not only to the fragmentation of $N_2O$ [35] but to another NO source as well. The formation of $N_2$ and ammonia as gaseous products ($m/z$ = 28 and $m/z$ = 17 or $m/z$ = 16, respectively) may be attributed to the decomposition of **compound 4** [33], but the mass loss showed that the $[Co_2(NH_3)_4Mn_6O_{12}]$ oxide phase also lost ammonia. The intensity ratio of $m/z$ = 18 ($H_2O^+$) and $m/z$ = 17 ($OH^+$ and $NH_3^+$) in this decomposition step was <1, which confirmed the ammonia evolution. Therefore, the decomposition product of **compound 4** was $NH_3$, $NH_4Cl$, and $N_2$ [33]:

$$6\,[Co(NH_3)_5Cl]Cl_2 = 6CoCl_2 + 6NH_4Cl + N_2 + 22NH_3$$

This process was a solid-phase quasi-intramolecular redox reaction centered on $Co^{III}$ and the ammonia ligand. Ammonium chloride (HCl) could not be detected directly due to its fast reaction with the capillary column organic film layer. The reaction heat (Figure S13) was found to be $-102.3$ and $-121.5$ kJ/mol in $O_2$ and $N_2$, respectively. This shows that the oxygen gas had an influence on the reaction heat. A similar effect of the outer oxygen was observed in the thermal decomposition of $[Ag(NH_3)_2ClO_4]$ [36]. That was attributed to the forming of NO (its formation was endothermic) in a higher amount than in an inert atmosphere. The second decomposition step of **compound 3-Mn** showed that the

formation of NO ($m/z = 30$), and the presence of Co-Mn oxides can help the oxidation of ammonia with the outer oxygen into NO.

In the second decomposition step of **compound 3-Mn** (~180 °C), the previously formed $NH_4NO_3$, $[Co(NH_3)_5Cl]Cl_2$ and $\{Co_2(NH_3)_4Mn_6O_{12}\}$ phases decomposed in simultaneous processes. The thermal decomposition temperature of $[Co(NH_3)_5Cl]Cl_2$ (**compound 4**) with $N_2$ and $NH_3$ formation (175 °C) [34] agreed well with the peak temperature of the second decomposition step (181 °C). The decomposition reaction of **compound 4** may initiate the decomposition of both ammonium nitrate (catalyzed by the oxide phases present [37–39]) and $\{Co_2(NH_3)_4Mn_6O_{12}\}$ simultaneously.

The third decomposition step around 340 °C was the reaction of the cobalt(II) chloride and $[Co_2(NH_3)_4Mn_6O_{12}]$ intermediates with the formation of cobalt manganese oxide ($CoMn_2O_4$), which contained formally one $Co^{II}$ and two $Mn^{III}$ ions. Two manganese (IV) ions or two cobalt (III) ions can oxidize two chloride ions into chlorine according to the reaction:

$$CoCl_2 + \{Co_2Mn_6O_{12}\} = 3CoMn_2O_4 + Cl_2$$

The mass loss of the third decomposition step was slightly higher than the mass losses calculated for $1/3$ $Cl_2$ due to the release of the residual ammonia. Still, the overall mass loss agreed well with the $CoMn_2O_4$ formation. Some amount of chlorine ($m/z = 35$ $Cl^+$) was also detected, in the argon without $NO_2$ formation whereas in the air it was together with $NO_2$ formation (Figure S11). The decomposition process is summarized in Scheme 1. As Scheme 1 shows, the only solid final decomposition product was $CoMn_2O_4$. Thus, the temperature-controlled decomposition of **compound 3-Mn** is an easy way to prepare nanosized cobalt manganite with Co:M$n$ = 1:2 stoichiometry.

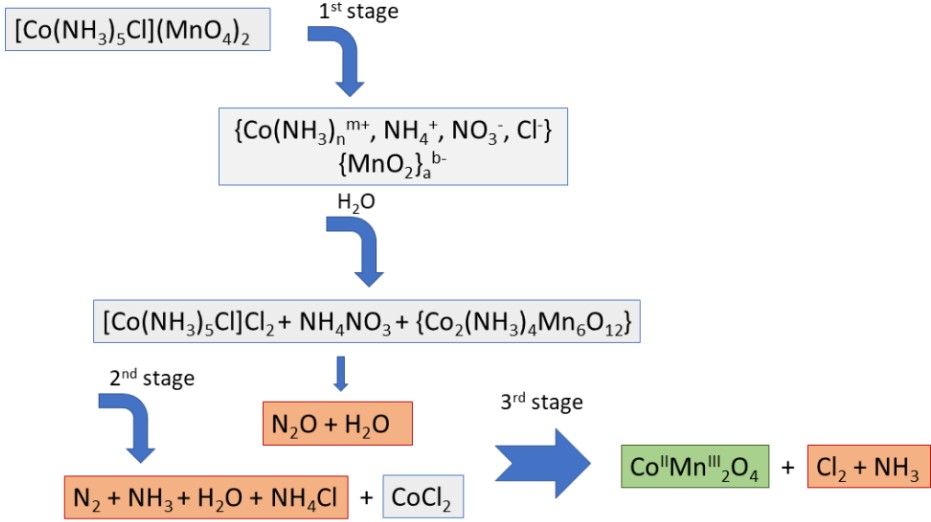

**Scheme 1.** Thermal decomposition of **compound 3-Mn**. Grey: solid intermediates; brown: gaseous decomposition products; green: solid decomposition product.

The stoichiometry of the third decomposition step requested the presence of two $Mn^{IV}$ or two $Co^{III}$ ions (or one $Mn^{III}$ and one $Co^{III}$), which could oxidize two chloride ions into chlorine. Consequently, 4 $Mn^{III}$ + *2Mn$^{IV}$* + 2$Co^{II}$, 5$Mn^{III}$ + *Mn$^{IV}$* + *Co$^{III}$* + $Co^{II}$, or 6$Mn^{III}$ + *2Co$^{III}$* ions (the oxidizing constituents are italicized) may have neutralized the negative charges of the framework oxygens in the $[Co_2(NH_3)_4Mn_6O_{12}]$. Thus, the $\{Co_2(NH_3)_4Mn_6O_{12}\}$ intermediate phase consisted of todorokite-like $\{Mn_4^{III}Mn^{IV}_2O_{12}\}_n^{4n-}$, $\{Mn_5^{III}Mn^{IV}O_{12}\}_n^{5n-}$ or $\{Mn^{III}_6O_{12}\}_n^{6n-}$ frameworks, which embedded $2 \times n$ ($Co^{II}$ and/or $Co^{III}$) cations in their tunnels, respectively. Ammonia ligands ($4 \times n$) were coordinated to cobalt ions. The two cobalt ions may have been intercalated in the tunnels as one $Co^{II,III}$ + one $Co^{II,III}$ $(NH_3)_4^{2+}$ ion, or two $[Co^{II,III}(NH_3)_2]^{2/3+}$ ions. $Co^{III}$ typically forms hexacoordinated, whereas $Co^{II}$ forms tetracoordinated ammonia complexes, thus, the most

probable formula contained a tetraamminecobalt (II) cation or ammonia-bridged Co (III) structural units, but the structure of this decomposition intermediate requires further detailed investigation.

The last decomposition reaction was influenced by oxygen, which means that oxygen defects ($CoMn_2O_{4+\delta}$) can be expected if the decomposition is completed in the presence of oxygen. Since the oxygen defects may play a role in the desired catalytic activity of Co-Mn oxides, the $CoMn_2O_4$ samples prepared in both an inert atmosphere and air were tested as photocatalysts in the UV degradation of Congo red.

### 2.6. Surface Characterization of the Thermal Decomposition Products of *Compound **3-Mn*** *and Their Photocatalytic Activity in Congo Red Degradation*

Twice were the three decomposition intermediates and the two final decomposition products prepared by isothermal heating of **compound 3-Mn** at 125, 150, 220 and 420 °C under $N_2$, and at 125, 150, 300 and 550 °C in air. The morphology of the end product (spinel-like) obtained at 420 °C ($N_2$) and at 550 °C (air) was similar and can be seen in Figure 11.

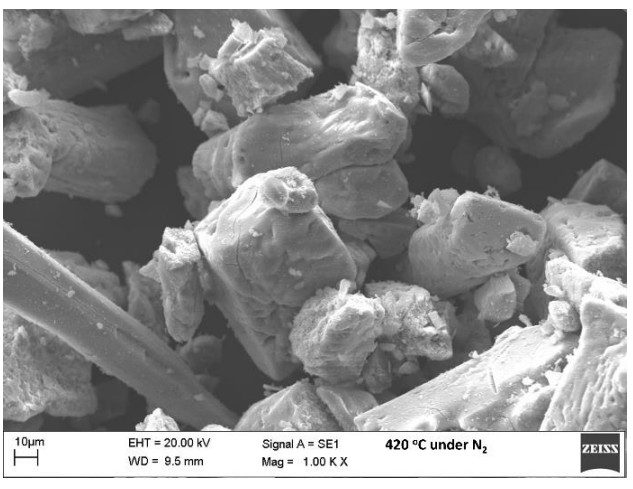

(**a**)

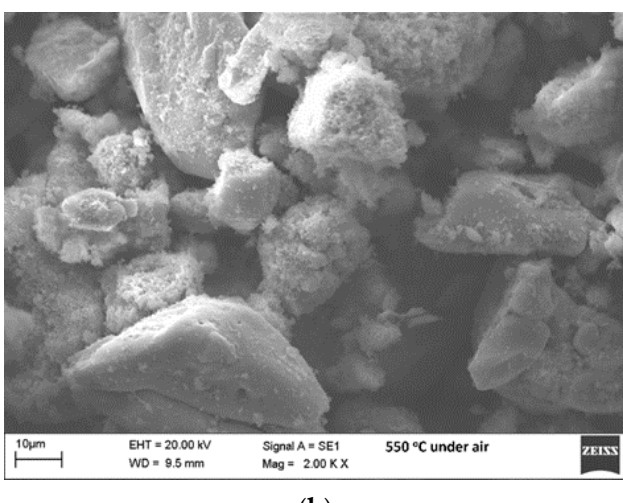

(**b**)

**Figure 11.** SEM images of the decomposition products formed at 420 °C under nitrogen (**a**) and at 550 °C under air (**b**).

The specific surface area of each intermediate and the final product was measured (Table 6).

**Table 6.** The specific surface area of intermediates formed at different temperatures under air and $N_2$.

| T °C (N2) | 125 | 150 | 220 | 420 |
|---|---|---|---|---|
| SSA ($m^2 \cdot g^{-1}$) | 130 | 118 | 176 | 32 |
| T °C (air) | 125 | 150 | 300 | 500 |
| SSA ($m^2 \cdot g^{-1}$) | 22 | 23 | 194 | 23 |

SSA: specific surface area.

The specific surface area of the decomposition intermediates formed under $N_2$ changed irregularly, and the lowest surface area (~32 $m^2 \cdot g^{-1}$) was found after crystallization and the sintering of the amorphous primary reaction products at 420 °C. Under an oxidizing atmosphere, there was an expansion in the bulk phase up to 300 °C, and then, the surface area decreased again due to crystallization and sintering at 550 °C (23 $m^2 \cdot g^{-1}$).

The first intermediate was obtained at 125 °C under both atmospheres and the end products were produced at 420 °C under nitrogen and at 550 °C in air. We checked

their potential as a photocatalyst in the degradation of organic dyes. Our findings are summarized in Figure 12 and the apparent decomposition rate for each intermediate can be seen in Table 7.

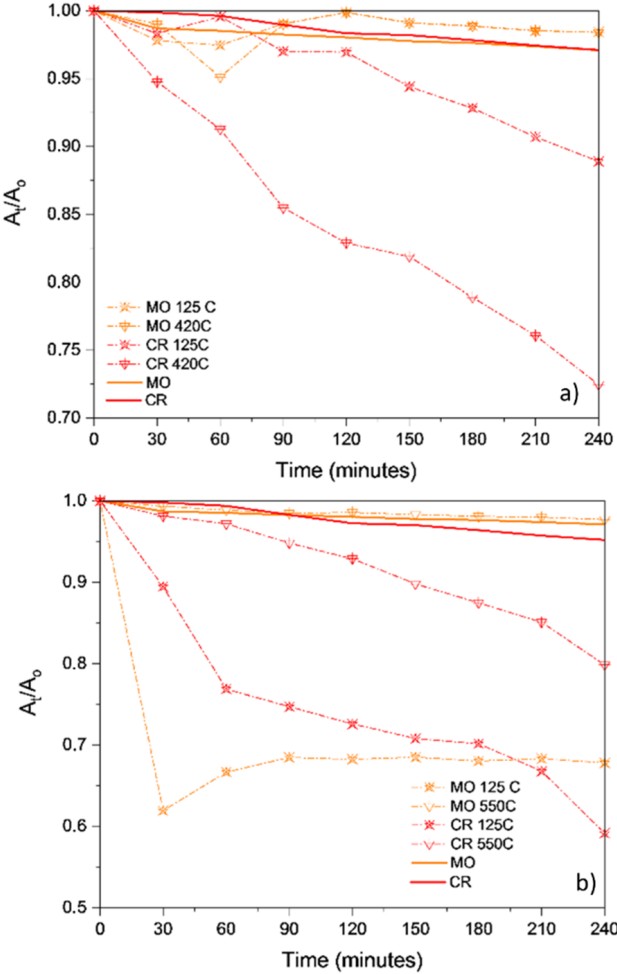

**Figure 12.** Photocatalysis of decomposition intermediates formed at various temperatures under nitrogen (**a**) and air (**b**) during the decomposition of Congo red.

**Table 7.** Photocatalytic parameters in the decomposition of methyl orange and Congo red with intermediates forming in different conditions.

| Substrate | pH | $K_{app}/10^{-4} \cdot min^{-1}$ | $R^2$ |
|---|---|---|---|
| Congo Red, $2 \cdot 10^{-5}$ M, without catalyst | 5.7 | 1.0 | 0.99 |
| Congo red, $2 \cdot 10^{-5}$ M, with **compound 3-Mn** at 125 °C in $N_2$ | 5.7 | 5.0 | 0.93 |
| Congo red, $2 \cdot 10^{-5}$ M, with **compound 3-Mn** at 125 °C in air | 5.7 | 18.0 | 0.89 |
| Congo red, $2 \cdot 10^{-5}$ M, with **compound 3-Mn** at 420 °C in $N_2$ | 5.7 | 13.0 | 0.98 |
| Congo red, $2 \cdot 10^{-5}$ M, with **compound 3-Mn** at 550 °C in air | 5.7 | 9.0 | 0.96 |

The photocatalyst candidates formed under an inert atmosphere and air during the thermal decomposition of **compound 3-Mn** showed activity in accelerating the decomposition rate of Congo red. The decomposition of methyl orange in our experiments did not reach any significant level. The end product ($CoMn_2O_4$) forming under $N_2$ had higher photoactivity (13 times faster degradation rate compared to the uncatalyzed process) in Congo red degradation by UV light than the intermediate [$Co_2(NH_3)_4Mn_6Mn_2O_{12}$]. The intermediate that formed in air at 125 °C accelerated the photodegradation of Congo red

18 times, whereas the final decomposition product (550 °C–air) resulted in only a 9-fold acceleration.

The differences in the chemical character and photocatalytic activity of samples prepared in an aerial and inert atmosphere can be attributed to the differences between the distribution and chemical environment of the low valence ($Co^{II}$, $Mn^{II}$) and oxidized ($Co^{III}$, $Mn^{III,IV}$) centers in the studied materials.

## 3. Materials and Methods

Chemical-grade basic cobalt carbonate, sodium permanganate (40% aq. soln.), ammonia and hydrochloric acid and chemicals containing the isotope ($D_2O$, DCl in $D_2O$, $ND_4Cl$, and $ND_3$ in $D_2O$ solution) were supplied by Deuton-X Ltd., Érd, Hungary.

**Synthesis of compound 4 [14]**

Basic cobalt carbonate—20 g of $CoCO_3 \cdot nCo(OH)_2 \cdot H_2O$—was dissolved in a 1:1 HCl solution. Then the solution was filtered and chilled. A solution of 250 mL of cc. ammonia and 50 g of $(NH_4)_2CO_3$ in 250 mL of distilled water was added. Then, the mixture was oxidized for three hours in a stream of air. Then, 150 g of $NH_4Cl$ was added to the previous solution and the mixture was evaporated until a syrup consistency was obtained. To drive off the $CO_2$, a dilute (10%) HCl solution was added. Then, to ensure the basicity of the reaction mixture, 10 mL of cc. ammonia was also added. The mixture was heated until all the solutions became clear, then 300 mL of cc. HCl was added, and the mixture was heated for 1 h. At the end of the process, the crude $[Co(NH_3)_5Cl]Cl_2$ was precipitated by cooling. The precipitate was filtered off, washed with 1:1 HCl until it was free of $NH_4Cl$, and then with alcohol until it was free of acid. To purify the crude salt, we dissolved 10 g of powdered crude $[Co(NH_3)_5Cl]Cl_2$ in 75 mL of distilled water and 50 mL of aqueous $NH_3$ (10%) in an Erlenmeyer flask covered with a watch glass under heating and stirring. Then, the deep-red solution was filtered and acidified with oxalic acid to be weakly acidic, and some additional $(NH_4)_2C_2O_4$ was added to complete the precipitation. The slurry was allowed to stand and the precipitated, $[Co(NH_3)_5(H_2O)]_2(C_2O_4)_3 \cdot 4H_2O$ was filtered off and washed with cold water and dried at room temperature. To obtain the pure **compound 4**, 20 g of $[Co(NH_3)_5(H_2O)]_2(C_2O_4)_3 \cdot 4H_2O$ was dissolved in 150 mL of diluted (2%) aq. ammonia solution in an ice bath, and the insoluble luteo oxalate, $[Co(NH_3)_6]_2(C_2O_4)_3 \cdot 4H_2O$, was filtered off. The filtrate was precipitated in an ice bath with the addition of a 10% aq. HCl solution. The [chloridopentaamminecobalt(III)] chloride (**compound 4**) was filtered off, washed successively with alcohol, absolute alcohol and ether, and dried in air.

**Synthesis of compound $[Co(NH_3)_5Cl](MnO_4)_2$ (compound 3-Mn)**

To prepare **compound 3-Mn**, we dissolved 1.0 g of $[Co(NH_3)_5Cl]Cl_2$ in 150 mL of distilled water and 5 mL of a $NaMnO_4$ solution was added. Then the mixture was heated to 60 °C and stirred for 15 min. Finally, the mixture was cooled to 1 °C. The precipitate that formed was filtered off and dried at room temperature in a desiccator containing CaO. The yield was 57%.

**Synthesis of deuterated compounds 4 and 3-Mn**

The deuterated **compound 4** could be prepared in three ways: (1) using anhydrous $CoCl_2$, $ND_4Cl$, and $ND_3$, and $D_2O$ according to the method described in [24]; (2) via deuteration of $[Co(NH_3)_5Cl]Cl_2$ dissolved in $D_2O$ and DCl and evaporated to dryness at room temperature in the presence of freshly prepared CaO; and (3) dissolving **compound 4** in $D_2O$ and freeze-drying the solution to obtain $[Co(ND_3)_5(D_2O)]Cl_3$, which was converted into deuterated **compound 4** by heating in a high vacuum at 80 °C, according to Sacconi's method [23]. All three ways gave the deuterated **compound 4**, but the deuteration of **compound 4** with $D_2O$ required a large amount of $D_2O$ because of the low solubility of **compound 4** in $H_2O$ ($D_2O$) (5.2 g/L at 25 °C [40]). The deuteration process was repeated 4 times, then the $[Co(ND_3)_5Cl]Cl_2$ was dissolved in a minimum quantity of $D_2O$ and reacted with 2 equiv. of $KMnO_4$ dissolved in $D_2O$ at 60 °C. Then the mixture was cooled, and the deuterated **compound 3-Mn** was isolated as described for the **non-deuterated 3-Mn**.

**Elemental Analysis**

The cobalt and manganese content of the sample was determined by atomic emission spectroscopy with a Spectro Genesis ICP-OES (SPECTRO Analytical Instruments GmbH, Kleve, Germany) spectrometer. A multielement standard solution (Merck Chemicals GmbH, Darmstadt, Germany) was used for calibration.

The ammonia content of the starting compound and the ammonia/ammonium-ion content of the decomposition intermediates were determined by gravimetry as $(NH_4)_2PtCl_6$ (ammonia was liberated by an alkaline treatment and boiling). Due to the redox by-reactions that appeared on heating of the **compound 3-M** which produced $NO_x$ gases, the absorption of the evolved ammonia in mineral acid solutions and titrating back the excess acid could not be used due to $HNO_2$ and $HNO_3$ formation.

The evolved ammonia/oxidized ammonia ratio was measured by the isotherm heating of the starting complex at a given temperature with the absorption of ammonia in the $H_2PtCl_6$ solution. The $NO_x$ and $N_2$ gases did not give a precipitate with $H_2PtCl_6$, whereas the ammonia formed insoluble orange-colored $(NH_4)_2PtCl_6$.

**Vibrational Spectroscopy**

The far-IR and FT-IR spectra of the crystalline **compound 3-Mn** and its deuterated form were recorded in ATR (attenuated total reflection) mode on a BioRad-Digilab FTS-30-FIR and a Bruker Alpha IR spectrometer for the 400–40 and 4000–400 $cm^{-1}$ range, respectively. Raman measurement (298 and 123 K) of **compound 3-Mn** and its deuterated form were performed on a Horiba Jobin-Yvon LabRAM microspectrometer with an external 785 nm diode laser source (~80 mW) coupled to an Olympus BX-40 optical microscope. In the low-temperature measurements, a Linkam THMS600 temperature-controlled microscope stage was used. The laser beam was focused on an objective of $20\times$. In order to avoid the thermal degradation of **compound 3-Mn,** we used a D0.6 (123 K) and a D2 (298 K) intensity filter to decrease the laser power to 25% and 1%, respectively. The confocal hole of 1000 μm and monochromators with 950 groove $mm^{-1}$ grating were used for light dispersion. The spectral range of 2000–100 $cm^{-1}$ was selected with a resolution of 4 $cm^{-1}$. The exposure times were 60 s to produce intensive peaks.

The UV-VIS DRS (UV-VIS diffuse reflectance spectrum) of **compound 3-Mn** was measured at room temperature with a Jasco V-670 UV–VIS spectrophotometer equipped with a NV-470 integrating sphere ($BaSO_4$ standard as reference).

**Scanning Electron Microscopy**

A JEOL JSM-5500LV scanning electron microscope was used in the experiments. The specimens were fixed on sample holders (Cu/Zn alloy with a carbon tape) and sputtered with a conductive layer (Au/Pd) for imaging.

**Powder X-ray diffractometry**

A Philips PW-1050 Bragg-Brentano parafocusing goniometer was used, equipped with a Cu cathode operated at 40 kV and 35 mA with a secondary beam graphite monochromator and a proportional counter. Scans were recorded in step mode. The diffraction patterns were evaluated with a full profile fitting technique.

**Single-crystal X-ray diffraction**

A dark violet needle of $[Co(NH_3)_5Cl](MnO_4)_2$ (**compound 3-Mn**) was mounted on a loop. Single crystal X-ray diffraction measurements were carried out on an Agilent Supernova diffractometer equipped with a dual-sealed X-ray tube source, a kappa goniometer and a position-sensitive detector at room temperature. An Mo source was used in a hemisphere of the reciprocal space up to the resolution of 0.65 Å. Data was collected with the CrysAlis1 software. The structure was solved and refined with the SHELX software package2 from Olex23. The atomic positions were determined by direct methods, while hydrogen atomic positions were calculated from assumed geometries. Hydrogen atoms were included in the structure factor calculations, but they were not refined. The isotropic displacement parameters of the hydrogen atoms were approximated from the U(eq) value of the atom they were bonded to.

**Thermal studies**

Thermal data were collected with a modified TGS-2 thermobalance (Perkin Elmer, USA) coupled to a HiQuad quadrupole mass spectrometer (Pfeiffer Vacuum, Germany). Approximately 1 mg samples were measured in a platinum sample pan. Decomposition was followed from ambient temperature to 500 °C at a 20 °C min$^{-1}$ heating rate in argon or air as the carrier gas (flow rate = 140 cm$^3$ min$^{-1}$). Selected significant ions between $m/z$ = 2–88 were monitored in the selected ion monitoring (SIM) mode.

The other set of thermal data were collected with a TA Instruments SDT Q600 thermal analyzer coupled to a Hiden Analytical HPR20/QIC mass spectrometer. Decomposition was followed from room temperature to 500 °C at a heating rate of 5 °C min$^{-1}$ in argon as the carrier gas (flow rate = 50 cm$^3$ min$^{-1}$) and at a heating rate of 2 °C min$^{-1}$ in air as the carrier gas (flow rate = 50 cm$^3$ min$^{-1}$). The sample holder and the reference were the same alumina crucible. A sample mass of ~1 mg was mixed with Al$_2$O$_3$ to avoid an explosive decomposition of the sample. Selected ions between $m/z$ = 1−83 were monitored in multiple ion detection modes (MIDs).

The non-isothermal DSC curves between −130 and 300 °C were recorded with a Perkin Elmer DSC 7 apparatus. The sample masses were between 3 and 5 mg and the samples were tested at a heating rate of 5 °C/min under a continuous nitrogen flow (20 cm$^3$ min$^{-1}$) in an unsealed aluminum pan.

## 4. Conclusions

1. An unknown cobalt(III) complex with chlorido and ammonia ligands, [Co(NH$_3$)$_5$Cl](MnO$_4$)$_2$ (**compound 3-Mn**), was synthesized. This complex is a precursor to nano-sized Co-manganite with Co:M$n$ = 1:2 stoichiometry and an average size of 16.8 nm.

2. The vibrational modes in **compound 3-Mn** were studied in detail and the overlapping bands were assigned via deuteration and low-temperature Raman measurements.

3. The structure of **compound 3-Mn** was revealed with single-crystal X-ray diffraction and the 3D-hydrogen bonds were evaluated.

4. The hydrogen bonds between the N-H hydrogen atoms and the Mn-O oxygen atoms acted as redox centers to initiate a solid-phase quasi-intramolecular redox reaction even at 120 °C involving the Co$^{III}$ centers and permanganate ion as the oxidants and ammonia as a reductant. An amorphous precursor of [Co(NH$_3$)$_5$Cl]Cl$_2$, NH$_4$NO$_3$, and a todorokite-like compound contains {Mn$_4$$^{III}$Mn$^{IV}$$_2$O$_{12}$}$_n$$^{4n-}$, {Mn$_5$$^{III}$Mn$^{IV}$O$_{12}$}$_n$$^{5n-}$ or {Mn$^{III}$$_6$O$_{12}$}$_n$$^{6n-}$ frameworks with 2 × n (Co$^{II}$ and/or Co$^{III}$) cations, and 4 × n ammonia ligand in their tunnels.

5. The decomposition intermediates formed at ~120 °C and decomposed on heating via a series of redox reactions with the formation of such redox products as Co$^{II}$Mn$^{III}$$_2$O$_4$ spinel, N$_2$, N$_2$O and Cl$_2$. 6) The only solid was the nanosized (16. 8 nm) CoMn$_2$O$_4$, with a Co:M$n$ = 1:2 stoichiometry.

6. The thermal decomposition reaction of **compound 3-Mn** consisted of a series of solid-phase quasi-intramolecular redox reactions consisting of various redox pairs such as permanganate ions and ammonia ligands, Co(III) and Mn(III), Co(III) and ammonia, or nitrate and ammonium ions.

7. The prepared CoMn$_2$O$_4$ spinel had photocatalytic activity in Congo red degradation with UV light. The activity of CoMn$_2$O$_4$ depended on its synthesis conditions such as temperature or atmosphere. Congo red degradation was 9 and 13 times faster in the presence of CoMn$_2$O$_4$ prepared at 520 °C (in air) or 450 °C (under N$_2$), respectively.

**Supplementary Materials:** The following supporting information can be downloaded at: https://www.mdpi.com/article/10.3390/inorganics10020018/s1, Figure S1: Comparison of experimental powder X-ray diffractogram and a calculated one (from data of single crystal measurement) of the **compound 3-Mn**; Figure S2: Correlation analysis for the internal and external vibrations of permanganate ion in **compound 3-Mn**; Figure S3: Correlation analysis for the internal and external vibrations of Co-Cl groups (molecular symmetry C$_{\infty v}$) in **compound 3-Mn**; Figure S4: Correlation analysis for the external vibrations of ammonia ligands in **compound 3-Mn**; Figure S5: Far-IR spectrum of

compound **3-Mn** in the range of 400–100 cm$^{-1}$ at room temperature; Figure S6: Far-IR spectrum of the deuterated **compound 3-Mn** in the range of 400–100 cm$^{-1}$ at room temperature; Figure S7: IR spectrum of the deuterated **compound 3-Mn** in the range of 4000–400 cm$^{-1}$ at room temperature; Figure S8: Raman spectra of the deuterated **compound 3-Mn** in the range of 1100–100 cm$^{-1}$ at 123 and 298 K; Figure S9: UV-VIS spectrum of **compound 3-Mn**; Figure S10: TG-DTG-DTA curves of **compound 3-Mn** in argon (a) and air (b) atmosphere; Figure S11: Selected TG-MS curves of **compound 3-Mn** in air (a) ($m/z = 35$, Cl$^+$; $m/z = 30$ (NO$^+$); $m/z = 46$, NO$_2^+$; $m/z = 18$, H$_2$O$^+$; $m/z = 17$ OH$^+$, NH$_3^+$) and argon (b) atmosphere ($m/z = 35$, Cl$^+$; $m/z = 30$ (NO$^+$); $m/z = 18$, H$_2$O$^+$; $m/z = 17$ OH$^+$, NH$_3^+$); Figure S12: Far-IR spectra of the thermal decomposition intermediates formed from **compound 3-Mn** in air (a) and inert (b) atmosphere; Figure S13: DSC analysis of **compound 3-Mn** in pure oxygen (a) and nitrogen (b); Table S1: Crystal data and structure refinement of **compound 3-M**; Table S2: Bond lengths (Å) and angles (°) in **compound 3-Mn**; Table S3: Analysis of potential hydrogen bonds and schemes with d(D . . . A) < R(D) + R(A) + 0.50, d(H . . . A) < R(H) + R(A)−0.12 Ang., D-H . . . A > 100.0 Deg in the crystal of [Co(NH$_3$)$_5$Cl](MnO$_4$)$_2$ (distances are given in Å).

**Author Contributions:** Conceptualization, F.P.F. and L.K.; methodology, F.P.F., K.A.B. and L.K.; software, F.P.F.; formal analysis, V.M.P.; investigation, F.P.F., É.K., Z.C., L.B., B.B.H., K.A.B. and A.F.; resources, I.M.S. and K.A.B.; data curation, V.M.P. and L.K.; writing—original draft preparation, F.P.F. and L.K.; writing—review and editing, F.P.F., K.A.B., V.M.P. and L.K.; visualization, L.B. and F.P.F.; supervision, L.K.; project administration, I.M.S.; funding acquisition, K.A.B., I.M.S. and L.K. All authors have read and agreed to the published version of the manuscript.

**Funding:** The research was supported by the European Union and the State of Hungary, co-financed by the European Regional Development Fund (VEKOP-2.3.2-16-2017-00013) (LK) and the ÚNKP-21-3 New National Excellence Program of the Ministry for Innovation and Technology from the source of the National Research, Development and Innovation Fund (KAB).

**Institutional Review Board Statement:** Not applicable.

**Informed Consent Statement:** Not applicable.

**Conflicts of Interest:** The authors declare no conflict of interest.

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
