# Peer review of "Multi-Centered Solid-Phase Quasi-Intramolecular Redox Reactions of [(Chlorido)Pentaamminecobalt(III)] Permanganate—An Easy Route to Prepare Phase Pure CoMn2O4 Spinel"

_inorganics, doi:10.3390/inorganics10020018_

Round 1
Reviewer 1 Report
I'd like to comment about the photocatalyst of Cong red. The authors should describe the different CoMn2O4 obtained by preparation. When I see Table 6, it looks like the different materials were obtained depending on under air or N2. How are these materials different? And also, what are the major factors for the photo-degradation rate of Cong red in Figure 11.
Author Response
Answers for reviewers
First of all, we would like to express our many thanks to the reviewers for improving our manuscript. All of the suggestions have been accepted and the text modified according to those.
Reviewer 1.
I'd like to comment about the photocatalyst of Cong red. The authors should describe the different CoMn2O4 obtained by preparation. When I see Table 6, it looks like the different materials were obtained depending on under air or N2. How are these materials different? And also, what are the major factors for the photo-degradation rate of Cong red in Figure 11.
Answer: Aerial oxygen can oxidize the low valence (CoII, MnIII) centers in the precursor of spinel material. It leads to the different distribution of CoIII, MnIII, or MnIV centers in the spinel octahedral and tetrahedral sites, which can cause the difference in catalytic properties. The samples prepared at different temperatures contains different valence distribution of cobalt and manganese in the different chemical environment. These differences in the chemical character and position within the crystalline lattices are the main factors (other factors as pH has been adjusted to be constant) that play a role in differences in catalytic activity in Congo red degradation.
A sentence about it has been inserted.
Reviewer 2 Report
The synthesis and single crystal structural characterization of [Co(NH3)5Cl](MnO4)2 complex as precursor of CoMn2O4 spinel is presented (also in deuterated form). The title compound was characterized by FT-IR, far-IR, low-temperature Raman and UV-Vis spectra. The N-H...O-Mn hydogen bonds act as redox centers to initiate a solid-phase "quasi-intramolecular" redox reaction,upon heating, which was extensively investigated with various techniques by the authors. The decomposition intermediates decompose on further heating via a series of redox reactions, forming a solid Co(II)Mn(III)2O4 spinel with an average size of 16.8 nm, and gaseous N2, N2O, and Cl. The obtained spinel has photocatalytic activity in Congo red degradation with UV light, with the activity strongly depending on the synthesis conditions.
The single crystal strcuture investigations, as well as the spectral and thermal investigation of the starting material and the intermediate products are presented in a careful and detailed manner (including also SEM and PXRD).
In order to emphasize the characterization of the final spinel product (in N2 and air) I recommend to present a series of PXRD pattern showing nanocrystalline, microcrystalline and theoertical line pattern of CoMn2O4 JCPDS 77-0471 (with same 2-Theta range) in the manuscript text itself (not in the SM) to allow non-specialized readers follow more easily.
Minor comments:
In Figures (manuscript and SM): (), [] or / in axis description were used: please harmonize.
Figure 1: Please use same type of atom labels in right and left part, i.e.: Co, Cl, Mn
Table 2: esd of unit cell volume missing
lines 111-113: esd of some bond parameters missing
PXRD pattern: Please replace Q by 2-Theta
Table S1 (SM): Separate max and min transmission values
Ref 2 in SM: cancel "50" before G.M. Sheldrick
Add reference for Scherrer equation
Author Response
First of all, we would like to express our many thanks to the reviewer for improving our manuscript. All of the suggestions have been accepted and the text modified according to those.
Reviewer 2.
The synthesis and single crystal structural characterization of [Co(NH3)5Cl](MnO4)2 complex as precursor of CoMn2O4 spinel is presented (also in deuterated form). The title compound was characterized by FT-IR, far-IR, low-temperature Raman and UV-Vis spectra. The N-H...O-Mn hydogen bonds act as redox centers to initiate a solid-phase "quasi-intramolecular" redox reaction, upon heating, which was extensively investigated with various techniques by the authors. The decomposition intermediates decompose on further heating via a series of redox reactions, forming a solid Co(II)Mn(III)2O4 spinel with an average size of 16.8 nm, and gaseous N2, N2O, and Cl. The obtained spinel has photocatalytic activity in Congo red degradation with UV light, with the activity strongly depending on the synthesis conditions.
The single crystal structure investigations, as well as the spectral and thermal investigation of the starting material and the intermediate products are presented in a careful and detailed manner (including also SEM and PXRD). In order to emphasize the characterization of the final spinel product (in N2 and air) I recommend to present a series of PXRD pattern showing nanocrystalline, microcrystalline and theoertical line pattern of CoMn2O4 JCPDS 77-0471 (with same 2-Theta range) in the manuscript text itself (not in the SM) to allow non-specialized readers follow more easily.
Answer: It has been inserted.
Minor comments:
In Figures (manuscript and SM): (), [] or / in axis description were used: please harmonize.
Answer: It has been done.
Figure 1: Please use same type of atom labels in right and left part, i.e.: Co, Cl, Mn
Answer: It has been done.
Table 2: esd of unit cell volume missing
Answer: It has been done
lines 111-113: esd of some bond parameters missing
Answer: It has been done
PXRD pattern: Please replace Q by 2-Theta
Answer: It has been done
Table S1 (SM): Separate max and min transmission values
Answer: It has been done.
Ref 2 in SM: cancel "50" before G.M. Sheldrick
Answer: It has been done.
Add reference for Scherrer equation
Answer: It has been inserted